# The Effectiveness and Safety of Pharmaceutical-Grade Cannabidiol in the Treatment of Mastocytosis-Associated Pain: A Pilot Study

**DOI:** 10.3390/biomedicines11020520

**Published:** 2023-02-10

**Authors:** Julien Rossignol, Séverine Hatton, Ashley Ridley, Olivier Hermine, Céline Greco

**Affiliations:** 1Department of Hematology, Necker-Enfants Malades Hospital, 75015 Paris, France; 2School of Medicine, Paris Cité University, 75006 Paris, France; 3Unité Mixte de Recherche (UMR) 1163, Institut National de la Santé et de la Recherche Médicale, 75015 Paris, France; 4Centre National de la Recherche Scientifique, 3 Rue Michel Ange, 75016 Paris, France; 5Reference Center for Mastocytosis (CEREMAST), 75015 Paris, France; 6Department of Pain and Palliative Care Unit, Necker-Enfants Malades Hospital, 75015 Paris, France

**Keywords:** mastocytosis, pain, IDO1, NMDAR, cannabidiol

## Abstract

Mastocytosis patients often experience a number of symptoms, including mastocytosis-associated pain that is difficult to manage due to resistance to usual antalgic treatments and/or the patient’s poor tolerance. Mastocytosis patients display significantly higher levels of indoleamine-2,3-dioxygenase-1 (IDO1) activity, leading to hyperactivation of the N-methyl-D-aspartate receptor. As cannabidiol (CBD) is known to inhibit IDO1′s enzymatic activity, we hypothesized that pharmaceutical-grade CBD is an effective treatment for mastocytosis-associated pain. Patients with non-advanced mastocytosis and refractory pain were eligible for inclusion in this observational pilot study. CBD was initiated at 50 mg/day and increased to a maximum of 900 mg/day. Pain was scored on a 0-to-10 numerical rating scale (NRS). A total of 44 patients were included over a 2-year period. The median dose of CBD prescribed was 300 mg/day (range: 50–900 mg/day). Elevated liver enzymes were observed in one patient. The mean ± standard deviation NRS pain score decreased significantly from 7.27 ± 1.35 before treatment to 3.78 ± 1.99 after 3 months of treatment (*p* < 0.0001). Fifteen patients (34%) were able to discontinue all their previous antalgic medications. CBD treatment might be a safe, effective treatment for mastocytosis-associated pain and its use requires confirmation in a randomized, controlled trial.

## 1. Introduction

Mastocytosis corresponds to a spectrum of diseases characterized by the infiltration of abnormal mast cells (MCs) into the skin and/or other organs. The 2016 World Health Organization (WHO) classification of mastocytosis includes three subgroups: mastocytosis in the skin (MIS, in which infiltration is limited to the skin), systemic mastocytosis (SM, in which tissues such as the bone marrow, gastrointestinal tract, and skeleton are infiltrated), and MC sarcoma [1]. SM is subdivided into five variants: indolent SM (ISM), smoldering SM (SSM), SM with an associated hematologic neoplasm, aggressive SM, and MC leukemia.

Patients with mastocytosis typically experience a number of symptoms related to MCs infiltration or activation. Surprisingly, pain has only recently been recognized as one of the most disabling symptoms of mastocytosis [2]. Patients frequently suffer from diffuse, chronic, moderate-to-severe pain (musculoskeletal pain, neuropathic pain, abdominal pain, and headache). The pain has a very negative impact on quality of life and often amplifies depressive symptoms, anxiety, sleep disturbance, severe asthenia, and feelings of social and professional isolation, thus exacerbating mastocytosis-related neuropsychiatric disorders [3,4,5]. The physiopathology of mastocytosis-associated pain is not well understood, and conventional antalgics are often poorly effective or badly tolerated, partly because the drugs themselves can activate MCs.

A previous study has shown that tryptophan metabolism is altered in patients with mastocytosis [6]. Tryptophan is an essential amino acid in mammals. In addition to its requirement in functional protein synthesis, tryptophan is metabolized in the kynurenine and serotonin pathways and thus produces various kynurenine metabolites (such as kynurenic acid, 3-hydroxykynurenine, quinolinic acid, and nicotinamide adenine dinucleotide) and a variety of neuroactive substances (such as serotonin and melatonin) [7]. Indoleamine-2,3-dioxygenase 1 (IDO1) and arylalkylamine N-acetyltransferase are the most important rate-limiting enzymes in the kynurenine and serotonin pathways, respectively [7]. It has been shown that patients with mastocytosis display significantly lower levels of serotonin (in the absence of hypoalbuminemia or malabsorption) and higher levels of IDO1 activity, which lead to higher levels of kynurenic acid and quinolinic acid [6]. The latter is an N-methyl-D-aspartate receptor (NMDAR) agonist, so higher levels of this metabolite result in hyperactivation of the NMDAR.

NMDARs are excitatory glutamatergic receptors involved in afferent nociceptive signal transmission [8]. In chronic pain states, prolonged nociceptive stimulation (i) causes activation and upregulation of the NMDARs at dorsal horn synapses, (ii) enhances and amplifies trafficking of pain signals to the brain, and (iii) thus leads to the presence of chronic persistent pain. In a recent clinical study, we found that anti-NMDAR drugs (such as ketamine, methadone, and memantine can effectively relieve pain in patients with mastocytosis [9]). However, the long-term use of these medications can produce adverse reactions, dependence, and addiction. Since these drugs can be misused, they should only be prescribed according to strict guidelines by primary care physicians who are aware of the associated risks. Furthermore, NMDARs are far downstream of the tryptophan degradation cascade, so targeting these receptors does not restore the serotonin/kynurenine balance.

Low levels of serotonin due to altered tryptophan metabolism and hyperactivation of the NMDARs by quinolinic acid caused by high levels of IDO1 could explain not only the refractory pain but also the anxiety and depression reported by mastocytosis patients.

It is well known that cannabinoids and the endocannabinoid system are involved in the regulation of anxiety and depression by modulating the serotonergic system in the central nervous system. More specifically, studies have shown that CBD can modulate serotoninergic signaling. It increases the availability of circulating tryptophan, which is the necessary precursor for neurotransmitter 5-hydroxytryptamine biosynthesis (5-HT; serotonin) [10]. CBD is also known to inhibit IDO1′s enzymatic activity. Consequently, it leads to rebalancing the serotonin/kynurenine ratio and to inhibiting the activation of NMDA receptors by decreasing quinolinic acid expression [11].

Given that chronic cannabidiol (CBD) use is well tolerated in patients with epilepsy, we hypothesized that pharmaceutical-grade CBD could be a safe, effective treatment for refractory mastocytosis-associated pain.

## 2. Methods

We conducted an observational, open-label, non-interventional, prospective pilot study at our university medical center’s pain care unit between 1 March 2020 and 1 March 2022. All the patients treated were enrolled in a prospective study sponsored by Association Française pour les Initiatives de Recherche sur le Mastocyte et les Mastocytoses (AFIRMM). Written informed consent was obtained from the patients for the publication of any potentially identifiable data included in this article. The AFIRMM study was approved by the local investigational review board (Comité de Protection des Personnes Ile-de-France, France; reference: 93-00) and was carried out in compliance with the principles of the Declaration of Helsinki. In line with the French legislation on non-interventional studies of routine clinical practice, the CBD study protocol was registered by Assistance Publique—Hopitaux de Paris under the following reference: MR004 2022 0516123418.

Patients were eligible for inclusion if they were aged 18 or over and suffered from mastocytosis (including MIS, ISM, and SSM), as defined in the WHO 2016 criteria. KIT mutations were assessed with droplet digital PCRs or Sanger sequencing.

Pain was measured on a numerical rating scale (NRS) ranging from 0 (“no pain”) to 10 (“worst imaginable pain”). Patients were included if they suffered from treatment-resistant chronic muscle, joint, bone, or abdominal mastocytosis-associated pain (defined as an NRS pain score of 6 or more for more than 3 months, despite treatment with standard antalgics) or poor tolerance of standard antalgics.

The impact of pain on the patient’s quality of life was assessed with the “Brief Pain Inventory Short Form”. BPI-Short Form is a 9-item self-administered questionnaire used to evaluate the severity of a patient’s pain and the impact of this pain on the patient’s daily functioning. The patient is asked to rate their worst, least, average, and current pain intensity, list current treatments and their perceived effectiveness, and rate the degree that pain interferes with general activity, mood, walking ability, normal work, relations with other persons, sleep, and enjoyment of life on a 10-point scale (0/10 “does not interfere” and 10/10 “completely interfere”).

The primary endpoint was the effectiveness of CBD in relieving pain after 1 month of treatment. The secondary endpoints were effectiveness at 3 months, the safety of CBD therapy, and a reduction in or withdrawal of other antalgics (opioids, ketamine, antidepressants, and antiepileptics).

Synthetic CBD was manufactured in Germany (Symrise, Holzminden, Germany) from limonene of orange peels. It is GMP-grade and nature-identical cannabidiol with no detectable traces of THC. The dose level of CBD (initially 50 mg/day) was increased by 50 mg every 7 days up to 150 mg/day (t.i.d.), then by 50 mg per dose every 7 days, to a maximum of 900 mg/day. Safety and effectiveness were evaluated during a teleconsultation; the dose level was decreased if adverse events occurred and increased if the NRS pain score had not decreased by more than 20% vs. baseline. At one and three months, patients were evaluated at our pain center; we checked their NRS diary and their consumption of other antalgics and took a blood sample for a liver enzyme assay. The initial and subsequent NRS pain scores were compared in a paired, two-tailed t-test. The threshold for statistical significance was set to *p* < 0.05. All statistical analyses were performed using Prism software (version 8.0, GraphPad Software, San Diego, CA, USA).

## 3. Results

A total of 44 eligible patients with mastocytosis starting CBD therapy were prospectively included. There were 37 women (84.1%), and the median age was 45.5. Thirty-nine patients (68.2%) had ISM, found had MIS (9.1%), and one had SSM (2.3%). All patients with ISM met the WHO 2016 major criterion for SM. The minor criteria, other characteristics, and the treatments at baseline are summarized for each patient in Table 1. The patient with SSM had a high MC burden (according to a bone marrow biopsy), a basal serum tryptase level above 200 ng/mL, and hepatosplenomegaly.

All patients completed the three-month course of treatment. The median dose level of CBD was 300 mg/day (mean: 373 mg/day, range: 50–900 mg/day). Elevated liver enzymes (ALT and AST < 5 upper limit of normal) were observed in a patient who was receiving more than 600 mg/day, although no clinical consequences were reported. The liver enzyme level normalized when the CBD treatment was temporarily discontinued for 14 days and did not rise when treatment was reinitiated.

The mean ± standard deviation NRS pain score decreased significantly from 7.27 ± 1.35 before treatment to 4.27 ± 2.29 after one month of CBD therapy (*p* < 0.0001) and 3.78 ± 1.99 after 3 months (M0 vs. M3: *p* < 0.0001; M1 vs. M3: *p* = 0.05) (Figure 1). For 22 patients (50.0%), the NRS pain score decreased by a factor of two or more. Fifteen of the forty-four patients (34.1%) were able to discontinue all their previous antalgic medications; of the remaining twenty-nine patients, four (13.8%) were able to reduce their antalgic intake by at least half. The median time for stopping or reducing treatment was 2 months.

All 44 patients completed the BPI-short form questionnaire at baseline and at 3 months. At 3 months of CBD treatment, 75% of patients were able to do more general activities (mean pain interference was 4.1/10 at 3 months versus 8.7/10 at baseline). In total, 56% saw their mood improve, felt less sad (mean pain interference 6.2/10 versus 8,6/10 at baseline), and showed sleep improvement, including fewer nocturnal awakenings- (mean pain interference 3.9/10 at 3 months versus 8.5/10 at baseline). Furthermore, 72% felt less limited in their physical efforts, including walking (mean pain interference 3.9/10 at 3 months versus 8.3/10 at baseline), and 45% felt less asthenic. Approximately 38% returned to work or to school, and 34% felt less social isolation (mean pain interference 6.6/10 at 3 months vs. 7.2/10 at baseline). Finally, 38% of patients felt less anxious and enjoyed their life more under CBD treatment (mean pain interference 5.1/10 at 3 months vs. 8.2/10 at baseline).

## 4. Discussion

Mastocytosis-associated pain is a disabling, difficult-to-treat symptom [2]. Patients most often present with the musculoskeletal pain/abdominal pain/headache triad, together with diffuse neuropathic pain in some cases. The patients’ functional capacity is markedly impacted: they may have to walk with a stick or a walker or use a wheelchair, they may have to give up work or studying, and their social and affective life becomes progressively poorer.

In the present study, a 3-month course of treatment with oral CBD was associated with an average three-point reduction in the NRS pain score. In our experience, this significant reduction is rarely produced by symptomatic treatment with antihistamines, antileukotrienes, or cromoglycate. In a study of omalizumab [12] for the treatment of mastocytosis-associated symptoms, pain was one of the least responsive endpoints. The effectiveness of CBD observed in the present study appears to be similar to that reported for the NMDAR inhibitors ketamine, memantine, and methadone. However, CBD has a better safety profile and is easier to prescribe because it is not classified as a narcotic. Thus, the only noticeable side effect reported for our study participants was a reversible elevation of liver enzyme levels in a patient taking more than 600 mg/d CBD. The elevation did not recur after the temporary discontinuation of the treatment. The pain reduction obtained with CBD is encouraging since pain has a major, negative impact on a patient’s quality of life and often amplifies neuropsychiatric disorders. Patients also reported improvement in their mood and sleep and a reduction of asthenic feelings and anxiety under CBD treatment.

We determined two mechanistic explanations for our present results. Firstly, a high level of IDO1 activity might shunt the available tryptophan toward the kynurenine pathway, increasing quinolinic acid synthesis and thereby lowering serotonin levels. This might account for the chronic, diffuse, moderate-to-severe pain and the depressive syndrome often reported by patients with mastocytosis [13]. As CBD reportedly inhibits IDO1′s enzymatic activity, the drug might restore the balance between the serotonin and kynurenine pathways, enhancing serotonin and decreasing quinolinic acid production, thus countering the activation of NMDARs [11]. Secondly, CBD might modulate the MCs’ immune response to inflammation. De Filippis et al. [14] showed that in a model of gut inflammation, CBD does not interact directly with CB1 or CB2 receptors on MCs but does interact directly with the peroxisome proliferator-activated receptor gamma and thus modulates the gut’s neuro-immune axis. More particularly (given that activated MCs recruit immune cells), they showed that CBD reduced the macrophage count and expression levels of tumor necrosis factor-alpha and MC chymase.

There is now an urgent need for translational research on the physiopathological mechanisms involved in the clinical improvement seen in our CBD-treated patients.

## 5. Conclusions

Treatment with pharmaceutical-grade CBD was associated with significant pain reduction over time in most of our patients with mastocytosis; hence, CBD might be a safe, effective treatment option for mastocytosis-associated pain. Prospective controlled trials are critically required to confirm these results on pain, deepen the preliminary data concerning neuropsychiatric symptoms improvement, and define its role in the armamentarium of mastocytosis symptomatic treatments.

## Figures and Tables

**Figure 1 biomedicines-11-00520-f001:**
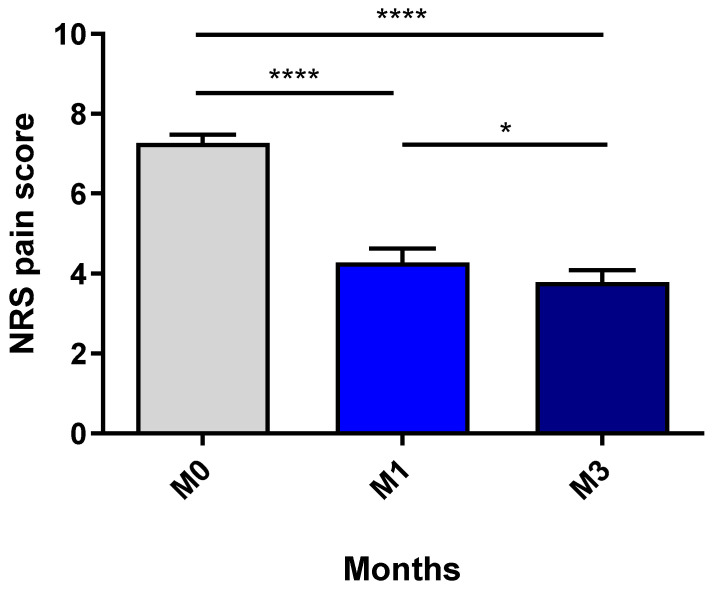
The mean NRS pain score before the initiation of CBD and after 1 and 3 months of treatment. NRS: numeric rate scale. M: months. NS: nonsignificant. Mean values were compared in a two-tailed, paired t-test. The threshold for statistical significance was set to * *p* < 0.05. (**** *p* < 0.0001).

**Table 1 biomedicines-11-00520-t001:** Characteristics of the patients and the mastocytosis, the numerical rating scale pain scores, the CBD dose level, and changes in other drug treatments.

Patient	Sex	Age (Years)	Type of Mastocytosis	*KIT* D816V Mutation	Serum Tryptase > 20 ng/mL	Atypical CD2/CD25 + MC	NRS Pain Score at M0	NRS Pain Score at M1	NRS Pain Score at M3	CBD Dose Level (mg/d)	Previous Antalgics
#01	F	33	ISM	Positive	Negative	Positive	7	3	3	300	no change
#02	F	41	ISM	Positive	Negative	Negative	6	4	4	600	no change
#03	M	63	ISM	Positive	Positive	Negative	9	2	2	300	discontinuation (ketamine)
#04	F	37	ISM	Positive	Negative	Positive	6	3	5	50	discontinuation (myorelaxant and opioids)
#05	F	37	ISM	Negative	Negative	Positive	9	2	1	50	no change
#06	M	59	ISM	Positive	Negative	Negative	8	5	6	150	no change
#07	F	44	ISM	Positive	Negative	Positive	6	4	3	450	discontinuation (myorelaxant and paracetamol)
#08	M	56	ISM	Negative	Negative	Positive	8	0	2	300	discontinuation (NSAIDs and antiepileptics)
#09	F	18	ISM	Negative	Positive	Negative	8	6	7	150	no change
#10	F	61	ISM	Positive	Negative	Negative	4	3	3	300	discontinuation (antidepressant, antiepileptics, and opioids)
#11	F	53	ISM	Negative	Positive	Negative	8	5	4	200	discontinuation (myorelaxant, anti-NMDA memantine, and ketamine)
#12	M	48	ISM	Positive	Negative	Negative	8	8	2	300	discontinuation (opioids)
#13	F	60	ISM	Positive	Positive	Positive	8, 5	8, 5	7, 5	150	discontinuation (antidepressant, antiepileptics, and opioids)
#14	F	33	ISM	Positive	Negative	Negative	10	4	5	100	no change
#15	F	34	ISM	Negative	Negative	Positive	5	5	4	300	two-fold reduction in opioids
#16	F	43	ISM	Positive	Positive	Negative	8	7	5	100	discontinuation (ketamine)
#17	F	60	SSM	Positive	Negative	Negative	7	5	5	250	no change
#18	F	46	ISM	Positive	Negative	Negative	6, 5	5	6	600	two-fold reduction in anti-depressant, opioids, and NSAIDs
#19	F	24	ISM	Negative	Negative	Positive	8	5	3	50	no change
#20	F	42	ISM	Positive	Negative	Positive	7, 5	4	6	250	discontinuation (opioids)
#21	F	72	ISM	Positive	Negative	Negative	6		4	150	discontinuation (opioids)
#22	F	51	ISM	Negative	Positive	Negative	8, 5	2	2	200	no change
#23	F	42	ISM	Positive	Negative	Negative	6, 5	3	2	450	no change
#24	M	21	ISM	Positive	Negative	Negative	7	3	2	750	no change
#25	F	45	ISM	Positive	Negative	Negative	8	4	3	300	discontinuation (NSAIDs and paracetamol)
#26	F	53	MIS	NA	NA	NA	8, 5	4	4	600	no change
#27	F	63	ISM	Negative	Positive	Positive	8	5	5	600	no change
#28	F	19	ISM	Negative	Negative	Positive	8	0	0	50	discontinuation (NSAID, paracetamol)
#29	F	29	ISM	Positive	Negative	Positive	7	6	6	200	no change
#30	F	52	ISM	Positive	Negative	Positive	6	6	6	600	discontinuation (ketamine)
#31	F	34	MIS	NA	NA	NA	8	7, 5	7, 5	450	no change
#32	F	66	ISM	Positive	Negative	Negative	7	5	5	300	two-fold reduction in antidepressant and myorelaxant use
#33	F	36	ISM	Positive	Negative	Negative	7	8	6	300	no change
#34	F	59	ISM	Positive	Negative	Positive	5	3	3	600	no change
#35	F	46	ISM	Negative	Positive	Positive	7	3	2	600	no change
#36	F	63	ISM	Positive	Negative	Positive	9	3	1	400	no change
#37	F	45	ISM	Positive	Negative	Positive	7	3	2	600	discontinuation (opioids)
#38	M	20	ISM	Positive	Positive	Positive	5	0	0	600	no change
#39	M	48	MIS	NA	NA	NA	6	2, 5	4	900	no change
#40	F	61	ISM	Positive	Negative	Positive	7	7	4	600	two-fold reduction in antiepileptic use
#41	F	68	ISM	Negative	Positive	Positive	10	10	5	450	no change
#42	F	25	MIS	NA	NA	NA	6	3	2	600	no change
#43	F	66	ISM	Positive	Negative	Negative	6	6	6, 5	900	no change
#44	F	44	ISM	Negative	Negative	Positive	9	1	1	300	no change

Mastocytosis was diagnosed as ISM, SSM or MIS, according to the WHO 2016 classification. All patients with SM met the major criterion. The patient with SSM had a high MC burden (on a bone marrow biopsy), a serum tryptase level above 200 ng/mL, and hepatosplenomegaly. F: female. M: male. NA: not available. NRS: numerical rating scale. M: months.

## Data Availability

No data additional to those reported in the manuscript were collected for this study.

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
