# Peer review of "The Effectiveness and Safety of Pharmaceutical-Grade Cannabidiol in the Treatment of Mastocytosis-Associated Pain: A Pilot Study"

_biomedicines, 2023, doi:10.3390/biomedicines11020520_

Round 1
Reviewer 1 Report
This is an interesting prospective pilot study providing evidence of CBD analgesia on patients with mastocytosis-associated pain.
I have no specific problem with this paper.
Just few points.
1 Authors stated that pain is a major issue for mastocytosis patients. However, patients seem more trouble more by anxiety, depression and personal look than by pain (Mesa et al., 2022; Spolak-Bobrik et al., 2022). As CBD can have also anxiolytic and antidepressant effects, these potential actions should be assessed (Berger et al., 2022; Stella, 2023).
2 Pain interference with ADLs should also be quantitated.
3 Although many patients continued the ongoing analgesics, a substantial percentage of patients reduce or discontinued therapy. Perhaps the authors may wish to graph the analgesic consumption.
Author Response
REVIEWER: This is an interesting prospective pilot study providing evidence of CBD analgesia on patients with mastocytosis-associated pain. I have no specific problem with this paper. Just few points.
AUTHORS: Thank’s for this positive evaluation. We have done our best to respond to the concerns of the reviewer.
- REVIEWER: Authors stated that pain is a major issue for mastocytosis patients. However, patients seem more trouble more by anxiety, depression and personal look than by pain (Mesa et al., 2022; Spolak-Bobrik et al., 2022). As CBD can have also anxiolytic and antidepressant effects, these potential actions should be assessed (Berger et al., 2022; Stella, 2023).
AUTHORS: Thank you for this question. You’re right. According to the literature, patients are also troubled by anxiety, depression and personal look. Low levels of serotonin due to altered tryptophan metabolism and hyperactivation of the NMDARs by quinolinic acid caused by high levels of IDO1 could explain not only the refractory pain but also the anxiety and depression reported by mastocytosis patients. It is well known that cannabinoids and the endocannabinoid system are involved in the regulation of anxiety and depression by modulating the serotonergic system in the central nervous system. More specifically, studies have shown that CBD can modulate serotoninergic signaling. It increases the availability of circulating tryptophan, which is the necessary precursor for neurotransmitter 5-hydroxytryptamine biosynthesis (5-HT; serotonin). CBD could therefore be an interesting and safer therapeutic option for improving refractory mastocytosis-related symptoms including pain and mood disorders.
These points has been added in the introduction section:
The following sentence (lines 46-49): “This pain is very often accompanied by a depressive syndrome, sleep disturbances, eating difficulties, and severe asthenia, which accentuate the patient’s social isolation”
Has been replaced by:
“The pain has a very negative impact on quality of life and often amplifies depressive symptoms, anxiety, sleep disturbance, severe asthenia and feelings of social and professional isolation, thus exacerbating mastocytosis-related neuropsychiatric disorders (Georgin-Lavialle S et al; 2016, Mesa et al; 2022, Spolak-Bobrik et al., 2022)”.
The following sentences (lines 79-92):
“Given that (i) chronic use of cannabidiol (CBD) is well tolerated in patients with epilepsy, (ii) CBD is known to inhibit IDO1’s enzymatic activity, and consequently (iii) the kynurenine signaling pathway leads to rebalancing of the serotonin/kynurenine ratio [8], we hypothesized that pharmaceutical-grade CBD could be a safe, effective treatment for refractory mastocytosis associated pain.”
Has been replaced by:
“Low levels of serotonin due to altered tryptophan metabolism and hyperactivation of the NMDARs by quinolinic acid caused by high levels of IDO1 could explain not only the refractory pain but also the anxiety and depression reported by mastocytosis patients.
It is well known that cannabinoids and the endocannabinoid system are involved in the regulation of anxiety and depression by modulating the serotonergic system in the central nervous system. More specifically, studies have shown that CBD can modulate serotoninergic signaling. It increases the availability of circulating tryptophan, which is the necessary precursor for neurotransmitter 5-hydroxytryptamine biosynthesis (5-HT; serotonin) (Florensa-Zanuy E et al; 2021). CBD is also known to inhibit IDO1’s enzymatic activity. Consequently it leads to rebalancing the serotonin/kynurenine ratio and to inhibiting the activation of NMDA receptors by decreasing quinolinic acid expression (Jenny et al, 2010).
Given that chronic use of cannabidiol (CBD) is well tolerated in patients with epilepsy, we hypothesized that pharmaceutical-grade CBD could be a safe, effective treatment for refractory mastocytosis associated pain.”
- REVIEWER: Pain interference with ADLs should also be quantitated.
AUTHORS: Thank you for your question. The impact of pain on the patient’s quality of life was assessed with the “Brief Pain Inventory Short Form” and we have added this information in this revised form.
These points have been added in the “methods” section and in the “results” section. However for future work, this aspect should be completed with more detailed scales (such as SF36, HAD and MC-QoL….) in another trial including mastocytosis patients for whom pain is not a predominant symptom, in order to have a better evaluation of the role of CBD on other symptoms than pain as mentioned now in the discussion.
In “methods”, the following sentences have been added (line 119-126):
“The impact of pain on the patient’s quality of life was assessed with the “Brief Pain Inventory Short Form”. BPI-short form is a 9-item self-administered questionnaire used to evaluate the severity of a patient's pain and the impact of this pain on the patient's daily functioning. The patient is asked to rate their worst, least, average, and current pain intensity, list current treatments and their perceived effectiveness, and rate the degree that pain interferes with general activity, mood, walking ability, normal work, relations with other persons, sleep, and enjoyment of life on a 10 point scale (0/10 “does not interfere” and 10/10 “completely interfere”).”
In “results”, the following sentences have been added (line 164-174):
All of the 44 patients completed the BPI-short form questionnaire at baseline and at 3 months. At 3 months of CBD treatment, 75% of patients were able to do more general activities (mean pain interference was 4,1/10 at 3 months versus 8,7/10 at baseline). 56% saw their mood improve, felt less sad (mean pain interference 6,2/10 versus 8,6/10 at baseline), and showed sleep improvement - including fewer nocturnal awakenings- (mean pain interference 3,9/10 at 3 months versus 8,5/10 at baseline). 72% felt less limited in their physical efforts including walking (mean pain interference 3,9/10 at 3 months versus 8,3/10 at baseline) and 45% felt less asthenic.38% returned to work or to school and 34% felt less social isolation (mean pain interference 6,6/10 at 3 months vs 7,2/10 at baseline). Finally, 38% of patients felt less anxious and enjoyed their life more under CBD treatment (mean pain interference 5,1/10 at 3 months vs 8,2/10 at baseline).
In “discussion”, the following sentence has been added (line 207-209 ):
« Patients reported also improvement of their mood and sleep, a reduction of asthenic feeling and anxiety under CBD treatment.”
and the following sentence (line 216):
“As CBD reportedly inhibits IDO1’s enzymatic activity, the drug might restore the balance between the serotonin and kynurenine pathways, decreasing quinolinic acid production, and thus counter the activation of NMDARs”
Has been modified:
“As CBD reportedly inhibits IDO1’s enzymatic activity, the drug might restore the balance between the serotonin and kynurenine pathways, enhancing serotonin and decreasing quinolinic acid production, and thus counter the activation of NMDARs”
In “conclusion”, the following sentence (line 230-231):
“Prospective controlled trials are critically needed to confirm these results and define its role in the armamentarium of mastocytosis symptomatic treatments.”
Has been modified:
“Prospective controlled trials are critically needed to confirm these results on pain, deepen the preliminary data concerning neuropsychiatric symptoms improvement and define its role in the armamentarium of mastocytosis symptomatic treatments”.
3.REVIEWER: Although many patients continued the ongoing analgesics, a substantial percentage of patients reduced or discontinued therapy. Perhaps the authors may wish to graph the analgesic consumption.
AUTHORS: Polytherapy is very common among this population. Most patients were simultaneously under opiods and other pain killers, anti-inflammatories, anti-depressants, anticonvulsants, muscle relaxers, etc. Due to the large number of medications and varying doses, we were unable to produce a clear and informative graph. The median time for stopping or reducing treatment was 2 months.
In “results”, the following sentence has been added (line 162-163):
“The median time for stopping or reducing treatment was 2 months”.
Reviewer 2 Report
In this paper entitled " The effectiveness and safety of pharmaceutical-grade canna- 2 bidiol in the treatment of mastocytosis-associated pain: a pilot 3 study “, the authors reported that treatment with pharmaceutical-grade cannabidiol (CBD) was associated with a significant reduction over time in pain in most of their patients with mastocytosis.
Given the absence of treatment for mastocytosis, cannabidiol treatment appears to be the most effective for mastocytosis associated pain.
The data is potentially informative and the paper deserves publication.
Author Response
REVIEWER 2: In this paper entitled " The effectiveness and safety of pharmaceutical-grade canna- 2 bidiol in the treatment of mastocytosis-associated pain: a pilot 3 study “, the authors reported that treatment with pharmaceutical-grade cannabidiol (CBD) was associated with a significant reduction over time in pain in most of their patients with mastocytosis. Given the absence of treatment for mastocytosis, cannabidiol treatment appears to be the most effective for mastocytosis associated pain. The data is potentially informative and the paper deserves publication.
AUTHORS: Thank’s for this very nice evaluation and for encouraging us in our effort to relieve pain in mastocytosis patients.